# Persistence of SARS-CoV-2 IgG Antibody Response among South African Adults: A Prospective Cohort Study

**DOI:** 10.3390/vaccines11061068

**Published:** 2023-06-06

**Authors:** Oladele Vincent Adeniyi, Oyewole Christopher Durojaiye, Charity Masilela

**Affiliations:** 1Department of Family Medicine, Cecilia Makiwane Hospital/Walter Sisulu University, East London 5200, South Africa; 2Department of Infection and Tropical Medicine, Royal Hallamshire Hospital, Sheffield Teaching Hospitals NHS Foundation Trust, Sheffield S10 2JF, UK; c.durojaiye@nhs.net; 3Department of Biochemistry, North-West University, Mahikeng 2745, South Africa; 43946194@nwu.ac.za

**Keywords:** SARS-CoV-2, COVID-19, humoral immune response, persistence, South Africa

## Abstract

This study assesses the durability of severe acute respiratory coronavirus-2 (SARS-CoV-2) anti-nucleocapsid (anti-N) immunoglobulin G (IgG) after infection and examines its association with established risk factors among South African healthcare workers (HCWs). Blood samples were obtained from 390 HCWs with diagnosis of coronavirus disease 2019 (COVID-19) for assay of the SARS-CoV-2 anti-N IgG at two time points (Phase 1 and 2) between November 2020 and February 2021. Out of 390 HCWs with a COVID-19 diagnosis, 267 (68.5%) had detectable SARS-CoV-2 anti-N IgG antibodies at the end of Phase I. These antibodies persisted for 4–5 and 6–7 months in 76.4% and 16.1%, respectively. In the multivariate logistic regression model analysis, Black participants were more likely to sustain SARS-CoV-2 anti-N IgG for 4–5 months. However, participants who were HIV positive were less likely to sustain SARS-CoV-2 anti-N IgG antibodies for 4–5 months. In addition, individuals who were <45 years of age were more likely to sustain SARS-CoV-2 anti-N IgG for 6–7 months. Of the 202 HCWs selected for Phase 2, 116 participants (57.4%) had persistent SARS-CoV-2 anti-N IgG for an extended mean period of 223 days (7.5 months). Findings support the longevity of vaccine responses against SARS-CoV-2 in Black Africans.

## 1. Introduction

The 2019 coronavirus disease (COVID-19) pandemic caused by severe acute respiratory syndrome coronavirus 2 (SARS-CoV-2) has brought about unparalleled health, economic and social crises around the world. Although the COVID-19 infection and death rate per capita has been relatively low in sub-Saharan Africa, South Africa has been significantly affected by the pandemic, with a highly contagious viral variant (501Y.V2) identified in late 2020 [1,2]. As of 11 April 2023, South Africa has recorded over 4 million cases and 102,595 COVID-19-related deaths [3]. Nevertheless, it appears that SARS-CoV-2 is here for the long term and could likely follow the path of most respiratory viruses, such as influenza and other human coronaviruses, which have a relatively predictable seasonal pattern [4,5,6]. Therefore, implementing effective control measures will require an understanding of the extent of natural immunity after infection. However, many questions remain about the durability of the immune response to SARS-CoV-2 infection. Additionally, the effect of established risk factors on the longevity of the humoral immune response remains a subject of intense investigation.

Ageing and chronic comorbidities have been identified as key risk factors for SARS-CoV-2 infection [7]. Furthermore, diabetes mellitus (DM), hypertension, obesity and cardiovascular diseases have been identified as the most predominant comorbidities among individuals infected with SARS-CoV-2 [7,8,9,10]. It has been shown that these comorbidities are risk factors for poor outcome and mortality in individuals infected with SARS-CoV-2 [8,10,11]. The mechanism in which these comorbidities worsen COVID-19 outcomes may involve immune and inflammatory responses, modulated by hyperglycaemia and the renin–angiotensin system [12,13,14,15].

Antibodies play a crucial role in neutralising viruses and preventing reinfection [16]. Upon infection, immunoglobulin M (IgM) antibodies are produced and start to decrease by the third week, while immunoglobulin G (IgG) antibodies are likely to remain detectable after infection [17]. For the human coronaviruses that cause the common cold, IgG antibodies last for at least a year after infection. In Middle East Respiratory Syndrome (MERS-CoV) and severe acute respiratory syndrome coronavirus-1 (SARS-CoV-1), IgG antibodies can be detected for years following infection [18]. Bichara et al. [19] reported a high incidence of loss of SARS-CoV-2 IgG antibodies within three months following COVID-19 diagnosis. On the other hand, other studies have reported a short- to medium-term protection against reinfection with similar levels of prevention of symptomatic infection as currently available vaccines [20,21].

Genetic variability in the immune system, more specifically in the human leukocyte antigen (HLA), may influence the strength of immune responses to SARS-CoV-2 [22,23]. The variations in the HLA system are linked to ethnicity and geographical location primarily due to pathogen-driven selection and past interbreeding with archaic human lineages [24,25]. The duration of COVID-19 immunity after a natural infection may also vary with age, body mass index (BMI), comorbidities and severity of infection [26,27]. Similar studies from USA revealed that sex, age, the duration of symptoms and disease severity were correlated with the loss and persistence of serum IgG levels among individuals who have recovered from COVID-19 [25,28].

Our current understanding of the durability of SARS-CoV-2 antibodies is limited by small sample sizes and the lack of data generated from African countries. Thus, this study investigated the durability of the humoral immune response (SARS-CoV-2 anti-N IgG antibody) to primary COVID-19 infection, and its association with age, sex, ethnicity and co-morbidities in a cohort of health care workers (HCWs) in South Africa. The findings from this study might improve our understanding of the humoral immune response to SARS-CoV-2 among the African population.

## 2. Materials and Methods

### 2.1. Study Design and Settings

This prospective cohort study was nested within the Eastern Cape Healthcare Workers Acquisition of the SARS-CoV-2 (ECHAS) study. The study aimed to determine the cumulative incidence of SARS-CoV-2 infection in a cohort HCWs in the Eastern Cape, South Africa [29]. The full methodology of the study had been published previously [30]. Briefly, this study was conducted in two teaching hospitals (Frere and Cecilia Makiwane Hospitals) located in the central region of the Eastern Cape Province of South Africa, which serve a combined population of about three million people residing in four district municipalities (Buffalo City Metropolitan, Amathole, Joe Gqabi and Chris Hani districts). In addition, the two hospitals are affiliated with the Walter Sisulu University for the training of both undergraduate and postgraduate students.

### 2.2. Participants and Procedure

Participants in this nested cohort study were recruited in two phases before the COVID-19 vaccine roll out in South Africa. In the first phase (Phase 1), we selected all the HCWs workers who self-reported a diagnosis of COVID-19 and were confirmed by a positive SARS-CoV-2 reverse transcription-polymerase chain reaction (RT-PCR) result (*n* = 390). Between November and December 2020, a trained research nurse collected 5 mL of venous blood from each participant for SARS-CoV-2 anti-N IgG antibody testing using aseptic technique. Participants who had positive for results for SARS-CoV-2 anti-N IgG antibodies were considered for follow-up after two months.

In the Phase 2, participants with persistent SARS-CoV-2 anti-N IgG antibody at the end of Phase 1 were invited telephonically for the follow-up study, and dates and time were agreed with the research team. A trained research nurse collected another 5 mL of venous blood for SARS-CoV-2 anti-N IgG antibody testing using aseptic technique. In the event of a difficult venepuncture, one of the doctors belonging to the research team collected the blood sample. Participants also answered questions on whether they had been diagnosed with COVID-19 within the previous two months.

### 2.3. Laboratory Testing

All venous blood samples were tested by the National Health Laboratory Service (NHLS) for IgG antibodies against the SARS-CoV-2 nucleocapsid protein in accordance with standard protocols. Collected serum samples were analysed using the Abbott SARS-CoV-2 IgG assay on an Abbott ARCHITECT i1000SR instrument, according to the manufacturer’s specifications. Briefly, this is a chemiluminescent microparticle-based immunoassay for the qualitative detection of IgG against the SARS-CoV-2 nucleocapsid protein, where the quality of the IgG present is reflected by the strength of response in relative light units. The relative units are further compared with a calibrator to determine the calculated index for a sample (with positive at 1.4 or greater). This assay has a sensitivity of 100% at 17 days after symptom onset and 13 days after PCR positivity and a specificity of 99.9% from 1020 pre-COVID-19 serum specimens [31].

All results were recorded on the Research Electronic Data Capture (REDCap) of the South African Medical Research Council server. This database houses the baseline dataset of the ECHAS study by assigning unique identifying numbers to encode the participants’ personal information (names and dates of births) but also allowing for the linkage of the follow-up data. This process ensures the confidentiality and privacy of medical information.

### 2.4. Main Outcome Measures

The main outcome measurement was the persistence of SARS-CoV-2 anti-N IgG antibodies at the end of Phase 1 (categorised as; 1–3 months, 4–5 months, and 6–7 months) and Phase 2 (7–8 months) following natural infection. In addition, factors associated with persistence of IgG antibodies were examined by extracting participants’ demographic and clinical characteristics from the ECHAS database.

The following explanatory variables were extracted from the database: age, gender, ethnicity, place of residence, highest level of education, smoking status and certain comorbidities (such as human immunodeficiency virus (HIV), hypertension, chronic kidney disease, DM, obesity and other cardiovascular diseases (CVDs)). These comorbid conditions have been shown to be associated with more severe disease or mortality [32,33,34,35,36]. In addition, the severity of COVID-19 disease was coded as severe if HCWs received oxygen or were admitted to the hospital.

### 2.5. Data Analysis

Data were directly exported from REDCap into the IBM SPSS Version 27.0 software (IBM SPSS, Chicago, IL, USA) and checked for accuracy. Descriptive statistics were used to summarize the demographic characteristics of the study participants and they were presented as counts and frequencies for categorical data and means for continuous data. Given the variable intervals between the RT-PCR-confirmed infections among the 390 participants and the first assay of SARS-CoV-2 anti-N IgG antibodies (Phase 1), the durability of these antibodies was assessed and categorised as follows: 1–3 months, 4–5 months and 6–7 months. The mean durability of SARS-CoV-2 anti-N IgG was also reported. Similarly, the extended durability of the SARS-CoV-2 anti-N IgG was tested at 7–8 months among 202 participants (Phase 2) who had persistent antibodies at the end of Phase 1.

We performed a chi-square test to assess the association between the baseline characteristics and the persistence of SARS-CoV-2 anti-N IgG antibodies at 4–5 months and 6–7 months in Phase 1 and the extended duration of 7–8 months in Phase 2. We excluded the 1–3 months from further analysis due to the small number of participants in this category (*n* = 31). Subsequently, we performed multivariate logistic regression model analysis fitted with a 95% confidence interval (95% CI) after controlling for other covariates to assess the independent association of participants’ characteristics and the persistence of SARS-CoV-2 IgG antibodies at 4–5 months, 6–7 months and the extended duration of 7–8 months. A two-tailed *p* value of less than 0.05 was considered statistically significant.

## 3. Results

### 3.1. Characteristics of the Participants (Phase 1)

A total of 390 HCWs who had tested RT-PCR positive for COVID-19 during the first wave of infection were enrolled in this study. The cohort comprises 332 female (85.12%) and 325 Black participants (83.33%). Most of the study participants (72.82%) were classified as obese (BMI ≥ 30 kg/m^2^), had tertiary education (73.33%) and had never smoked cigarettes (94.87%) (Table 1).

### 3.2. SARS-CoV-2 Anti-N IgG Antibody Durability

Out of 390 HCWs who had tested RT-PCR positive for COVID-19 at baseline, 68.46% (*n* = 267) had detectable SARS-CoV-2 anti-N IgG antibodies at the end of Phase 1 (Figure 1). These antibodies persisted for 1–3, 4–5 and 6–7 months in 7.49% (*n* = 20), 76.40% (*n* = 204) and 16.10% (*n* = 43), respectively. Both active smokers (7 out of 11) and former smokers (6 out of 9) were able to sustain the SARS-CoV-2 anti-N IgG antibodies for 4–5 months. The mean durability of SARS-CoV-2 anti-N IgG antibodies was 143 days (4.67 months).

### 3.3. Factors Associated with Antibody Durability

In the chi-square analysis, age (*p* < 0.001), hypertension (*p* = 0.013) and obesity (*p* = 0.027) showed a significant association with SARS-CoV-2 anti-N IgG antibody durability for 4–5 months. Besides obesity (*p* < 0.001), there were no associations established between the selected variables and SARS-CoV-2 anti-N IgG antibody durability at 6–7 months (Table 2).

In the multivariate logistic regression model analysis, Black participants were more likely to sustain SARS-CoV-2 IgG antibodies for 4–5 months (adjusted odds ratio (AOR), 7.85; 95% confidence interval [CI] 1.49–41.14; *p* = 0.02). Participants who were human immunodeficiency virus (HIV) positive were less likely to sustain SARS-CoV-2 anti-N IgG antibodies for 4–5 months (AOR = 0.16; 95% CI 0.03–0.76; *p* = 0.02). In addition, individuals who were <45 years of age were more likely to sustain SARS-CoV-2 IgG antibodies for 6–7 months (AOR, 3.38; 95% CI, 1.02–11.22; *p* = 0.046). Other selected variables were not associated with SARS-CoV-2 anti-N IgG antibody persistence (Table 3).

### 3.4. Phase 2: Extended Antibody Durability (7–8 Months)

In the Phase 2, 202 HCWs out of 267 who exhibited SARS-CoV-2 anti-N IgG antibody seropositivity during the Phase 1 participated. The persistence of the SARS-CoV-2 anti-N IgG antibody occurred in 57.43% of the participants (*n* = 116) (Figure 2). None of the active smokers and formers smokers sustained SARS-CoV-2 anti-N IgG antibody for an extended period.

The mean extended durability of SARS-CoV-2 anti-N IgG antibodies was 223 days (7.46 months). Both the Chi-square (Table 4) and multivariate logistic regression model analysis (Table 5) showed that the selected variables had no significant associations with the extended durability of SARS-CoV-2 anti-N IgG antibodies.

## 4. Discussion

One of the key questions in predicting the course of the COVID-19 pandemic and future interventions is how well and how long protective immunity might last after infection or vaccination. However, the exact duration of SARS-CoV-2 immunity after a natural infection is not completely understood. In this study, we investigated the durability of the humoral immune response (SARS-CoV-2 anti-N IgG antibodies) to primary infection, and its association with factors such as age, sex, ethnicity and co-morbidities among South African HCWs.

In our study, the mean durability of SARS-CoV-2 anti-N IgG antibodies was approximately five months. In some individuals, the antibody response persisted for over seven months. Similar to other studies, we identified a proportion of individuals with RT-PCR-confirmed SARS-CoV-2 who did not have detectable antibodies [37,38,39,40]. The reasons why some participants who were previously diagnosed with COVID-19 did not develop antibodies against the virus are unclear but may be related to asymptomatic infection or milder disease [39,40]. Nonetheless, our findings add to the growing evidence that SARS-CoV-2 anti-N IgG antibodies can persist for at least three months, and possibly longer, after infection. Of note, this is the first study to explore and report on the durability of SARS-CoV-2 anti-N IgG antibodies after natural infection among South Africans.

People with HIV, particularly those with advanced disease, are at higher risk of severe complications resulting from SARS-CoV-2 infection independent of sex and age [41,42]. However, limited data exist on natural immunity after SARS-CoV-2 infection among this subgroup. In the current study, HIV-positive participants were more likely to be seronegative at 4–5 months. Similarly, Spinelli et al. [43] demonstrated that individuals with HIV had lower neutralising antibody titres and SARS-CoV-2 IgG concentrations after natural infection. These findings suggest that immunosuppressive conditions may influence immune response to SARS-CoV-2 and increase the risk of reinfection.

The ability to mount and maintain appropriate immune response in the presence of DM is important for preventing SARS-CoV-2 re-infection and mortality [44]. In this study, pre-existing DM was associated with a sustained anti-N IgG antibody response. The reason for this observation is unclear, although a previous study reported that DM was associated with higher IgG levels [45], while others reported that individuals with good glycaemic control had less severe COVID-19 pneumonia and lower mortality risk in comparison to those with poor glycaemic control [46,47]. Other studies revealed that hyperglycaemia has no effect on antibody response against SARS-CoV-2 [47]. Among patients with DM, it has been suggested that SARS-CoV-2 infection impairs glucose homeostasis and metabolism mainly due to cytokine storm and the direct injury of pancreatic beta cells [48]. Although more studies are needed to examine the SARS-CoV-2 IgG antibody responses in this subpopulation, our findings create optimism regarding the durability of SARS-CoV-2 vaccines’ responses among this subgroup.

The effects of ethnicity/race, age and disease severity on SARS-CoV-2 IgG antibody longevity remain largely unknown. Unlike similar studies [28,39,49], we did not identify age or disease severity to be associated with a longer lasting immune response. However, Black participants were more likely to have detectable antibodies at 4–5 months. We found no record of the longevity of SARS-CoV-2 IgG antibody among Black individuals in any study. Even so, other studies reported that non-white races, including African Americans, may be associated with higher prevalence of antibodies following infection [39,49]. Furthermore, Smith et al. [50] reported a major difference in the degree and strength of the humoral immune response in relation to ethnicity, attributed to genetic and selected lifestyle factors. It is important to note that a large proportion of the participants in the current study were Black Africans. Therefore, these findings should be interpreted with caution. Future studies from South Africa should target other racial groups to gain a better understanding of the effect of race on humoral immune response against SARS-CoV-2 infection. 

Smoking is a common unhealthy habit that is prevalent world-wide [51], and it is a major risk factor for respiratory infection and disease progression [52,53]. A previous study revealed that smoking was not associated with the risk of SARS-CoV-2 infection [54]. On the other hand, other studies reported that smoking was associated with a high risk of COVID-19 [55] and that smokers have worse outcomes following SARS-CoV-2 infection [52,56]. Nonetheless, the role of smoking on the durability of SARS-CoV-2 anti-N IgG antibodies remains unknown. While a multivariate logistic regression analysis was not conducted to assess the effect of smoking on the durability of SARS-CoV-2 anti-N IgG antibodies due to the small number of smokers in this study, we observed that active smokers and former smokers were able to sustain the SARS-CoV-2 anti-N IgG antibodies for 4–5 months. However, neither of the active smokers nor former smokers were able to sustain their SARS-CoV-2 anti-N IgG antibodies beyond six months. Future studies should validate the observations made in this study by exploring the association between smoking and the durability of SARS-CoV-2 anti-N IgG antibodies.

The humoral immune response to SARS-CoV-2 includes antibodies against specific viral antigens, such as the nucleocapsid protein and spike (S) protein [57]. The S protein harbours the receptor binding domains (RBD) that allows the virus to bind and enter susceptible host cells [57,58]. Although most of the participants in our study developed detectable levels of SARS-CoV-2 anti-N IgG antibodies, the extent and duration of protection that these antibodies offer against SARS-CoV-2 reinfection is unclear. The development of antibodies may result in some level of protection against SARS-CoV-2 reinfection [40]. We were not able to assess the functional ability of the detected antibodies through a neutralisation test due to lack of required resources in our setting. Nevertheless, numerous studies have reported that people with SARS-CoV-2 IgG antibodies are less likely to be reinfected in the short-term than those who lack the antibodies [20,59,60]. The appearance of new SARS-CoV-2 variants calls for further studies to assess the extent of neutralising antibodies and protection from subsequent infection.

### Study Limitations

Our study is geographically specific and includes a relatively small number of HCWs, potentially limiting generalisability to other settings or populations. However, our findings are consistent with other studies based on the durability of SARS-CoV-2 anti-N IgG antibodies [61,62]. Additionally, the current study could not assess the durability of the humoral immune response in individuals with tuberculosis and other respiratory diseases due to their small number. More studies are needed to better elucidate the persistence of SARS-CoV-2 anti-N IgG antibodies in these sub-populations. In addition, we could not assess how SARS-CoV-2 anti-N IgG antibodies correlate with functionality and protection against reinfection.

## 5. Conclusions

Our study showed sustained humoral immunity in a cohort of African HCWs who recovered from COVID-19. The sustained antibody response is not affected by age, race/ethnicity, comorbidities or disease severity. Our findings support the longevity of vaccine responses against SARS-CoV-2 in Black Africans. However, periodic vaccinations with updated vaccines or boosters may be required, as they are for seasonal influenza.

## Figures and Tables

**Figure 1 vaccines-11-01068-f001:**
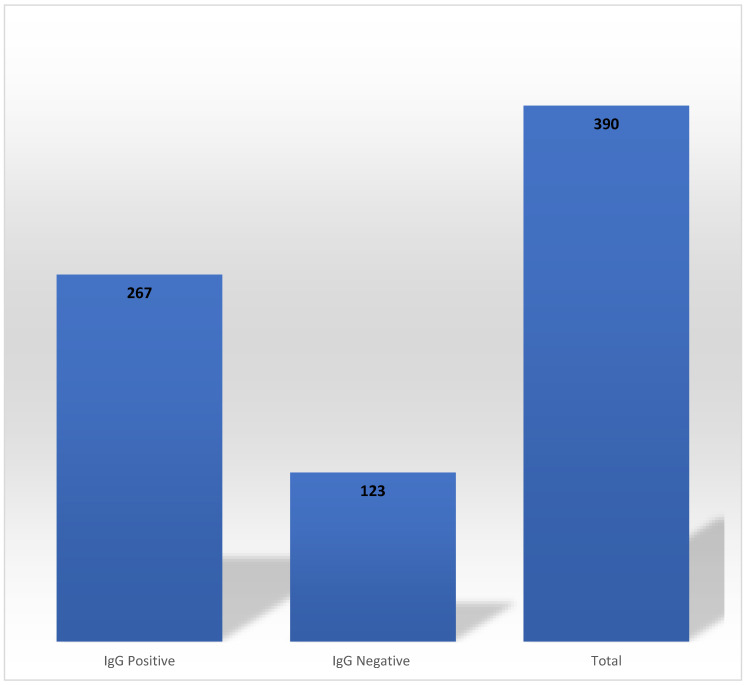
Persistence of SARS-CoV-2 anti-N IgG antibodies in Phase 1. SARS-CoV-2; severe acute respiratory syndrome coronavirus 2.

**Figure 2 vaccines-11-01068-f002:**
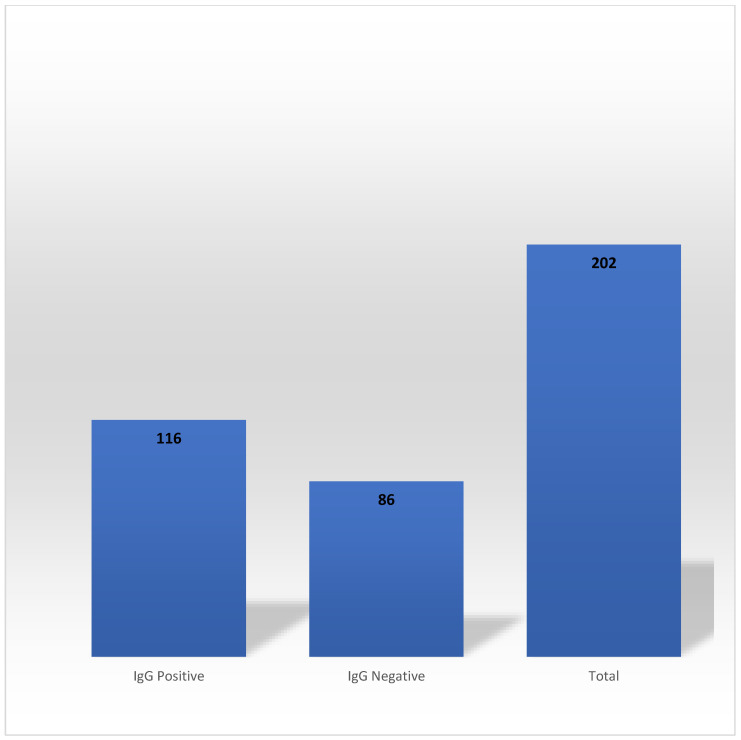
Persistence of SARS-CoV-2 anti-N IgG antibodies in Phase 2. SARS-CoV-2; severe acute respiratory syndrome coronavirus 2.

**Table 1 vaccines-11-01068-t001:** Baseline characteristics of the participants.

Variable	Frequency (*n* = 390)	Percentage (%)
Sex		
Males	58	14.9
Females	332	85.1
Age (Years)		
18–25	12	3.1
26–35	94	24.1
36–45	99	25.4
46–55	117	30.0
>55	68	17.4
BMI (kg/m^2^)		
<18.5	02	0.5
18.5–24.9	30	7.7
25.0–29.9	74	19.0
≥30	284	72.8
Ethnicity		
Black	325	83.3
White	23	5.9
Mixed-race	28	7.2
Other	14	3.6
Residential Area		
Urban	205	52.6
Suburban	185	47.4
Level of Education		
Tertiary	286	73.3
Secondary	104	26.7
Smoking Status		
Never smoked	370	94.9
Active smoker	11	2.8
Former smoker	09	2.3

BMI, body mass index.

**Table 2 vaccines-11-01068-t002:** Chi-square analysis of factors associated with SARS-CoV-2 IgG antibody persistence in Phase 1.

Variable	4–5 Months		6–7 Months	
	IgG Positive (*n* = 204)	IgG Negative (*n* = 86)	IgG Positive (*n* = 43)	IgG Negative (*n* = 26)
Ethnicity				
Black	171 (70.7)	71 (29.3)	39 (62.9)	23 (37.1)
White	12 (66.7)	6 (33.3)	2 (50.0)	2 (50.0)
Mixed-race	14 (66.7)	7 (33.3)	1 (50.0)	1 (50.0)
Other	7 (77.8)	2 (22.2)	1 (100.0)	0 (0.0)
*p*-values	0.918		0.801	
Age (Years)				
<45	105 (68.2)	49 (31.8)	29 (65.9)	15 (34.1)
≥45	99 (72.8)	37 (27.2)	14 (56.0)	11 (44.0)
*p*-values	<0.001		0.414	
Diabetes Mellitus				
Yes	15 (65.2)	8 (34.8)	8 (66.7)	4 (33.3)
No	189 (70.8)	78 (29.2)	35 (61.4)	22 (38.6)
*p*-values	0.575		0.732	
Hypertension				
Yes	50 (83.3)	10 (16.7)	12 (66.7)	6 (33.3)
No	154 (67.0)	76 (33.0)	31 (60.8)	20 (39.2)
*p*-values	0.013		0.658	
HIV				
Yes	13 (76.5)	4 (23.5)	05 (71.4)	2 (28.6)
No	191 (70.0)	82 (30.0)	38 (61.3)	24 (38.7)
*p*-values	0.569		0.600	
Other NCDs				
Yes	17 (89.5)	2 (10.5)	2 (50.0)	2 (50.0)
No	187 (69.0)	84 (31.0)	41 (66.1)	21 (33.9)
*p*-values	0.059		0.600	
Obesity				
Yes	157 (74.4)	54 (25.6)	37 (75.5)	12 (24.5)
No	47 (60.3)	31 (39.7)	6 (30.0)	14 (70.0)
*p*-values	0.027		<0.001	
Severity of infection				
Mild	185 (69.3)	82 (30.7)	36 (59.0)	25 (41.0)
Severe	19 (82.6)	4 (17.4)	7 (87.5)	1 (12.5)
*p*-values	0.180		0.118	

Data are presented as *n* (%) unless otherwise indicated. HIV, human immunodeficiency virus; NCDs, non-communicable diseases; SARS-CoV-2; severe acute respiratory syndrome coronavirus 2.

**Table 3 vaccines-11-01068-t003:** Multivariate logistic regression analysis of factors associated with SARS-CoV-2 IgG antibody persistence in Phase 1.

Variables	4–5 Months (*n* = 204)		6–7 Months (*n* = 43)	
	Adjusted Odds Ratios (95% CI)	*p*-Values	Adjusted Odds Ratios (95% CI)	*p*-Values
Ethnicity				
Black	7.85 (1.49–41.14)	0.02	8.66 (0.68–109.19)	0.09
White	5.26 (0.43–64.96)	0.19	5.81 (0.20–166.22)	0.30
Mixed-race	0.97 (1.55–6.14)	0.98	0.42 (0.02–10.86)	0.60
Other	Ref		Ref	
Age (Years)				
<45	1.62 (0.58–4.56)	0.36	3.38 (1.02–11.22)	0.046
≥45	Ref		Ref	
Diabetes Mellitus				
Yes	0.29 (0.06–1.39)	0.12	0.95 (0.17–5.21)	0.96
No	Ref		Ref	
Hypertension				
Yes	3.17 (0.75–13.39)	0.12	3.05 (0.62–14.98)	0.17
No	Ref		Ref	
HIV				
Yes	0.16 (0.03–0.76)	0.02	0.32 (0.06–1.82)	0.19
No	Ref		Ref	
Other NCDs				
Yes	0.85 (0.161–4.54)	0.86	0.42 (0.05–3.75)	0.44
No	Ref		Ref	
Obesity				
Yes	1.22 (0.36–4.14)	0.75	2.05 (0.45–9.38)	0.35
No	Ref		Ref	

CI, confidence interval; HIV, human immunodeficiency virus; NCDs, non-communicable diseases; SARS-CoV-2; severe acute respiratory syndrome coronavirus 2.

**Table 4 vaccines-11-01068-t004:** Chi-square analysis of factors associated with extended SARS-CoV-2 IgG antibody persistence.

Variable	Phase 2		*p*-Values
	IgG Positive (*n* = 116)	IgG Negative (*n* = 86)	
Ethnicity			0.99
Black	96 (57.5)	71 (42.5)	
White	6 (66.7)	3 (33.3)	
Mixed-race	10 (58.8)	7 (41.2)	
Other	6 (54.5)	5 (45.5)	
Age (Years)			0.48
<45	57 (60.0)	38 (40.0)	
≥45	59 (55.1)	48 (44.9)	
Diabetes Mellitus			0.97
Yes	12 (57.1)	9 (42.9)	
No	104 (57.5)	77 (42.5)	
Hypertension			0.20
Yes	35 (64.8)	19 (35.2)	
No	81 (54.7)	67 (45.3)	
HIV			0.45
Yes	10 (66.7)	5 (33.3)	
No	106 (56.7)	81 (43.3)	
Other NCDs			0.98
Yes	8 (57.1)	6 (42.9)	
No	108 (57.4)	80 (42.6)	
Obesity			0.54
Yes	91 (56.2)	71 (43.8)	
No	24 (61.5)	15 (38.5)	
Chronic Kidney Disease			0.83
Yes	1 (50.0)	1 (50.0)	
No	115 (57.5)	85 (42.5)	
Severity of Disease			0.43
Mild	104 (58.4)	74 (41.6)	
Severe	12 (50.0)	12 (50.0)	

Data are presented as *n* (%) unless otherwise indicated. HIV, human immunodeficiency virus; NCDs, non-communicable diseases; SARS-CoV-2; severe acute respiratory syndrome coronavirus 2.

**Table 5 vaccines-11-01068-t005:** Multivariate logistic regression analysis showing factors associated with extended SARS-CoV-2 IgG antibody persistence in Phase 2.

Variable	Adjusted Odds Ratios	*p*-Value
Ethnicity		
Black	1.27 (0.35–4.69)	0.71
White	0.97 (0.14–6.78)	0.98
Mixed-race	1.33 (0.268–6.57)	0.73
Other	Ref	
Age (Years)		
<45	1.29 (0.76–2.30)	0.39
≥45	Ref	
Nature of the Disease		
Severe	0.68 (0.26–1.82)	0.44
Mild	Ref	
Diabetes Mellitus		
Yes	1.06 (0.36–3.12)	0.91
No	Ref	
Hypertension		
Yes	1.6 (0.82–3.28)	0.16
No	Ref	
HIV		
Yes	1.55 (0.49–4.93)	0.45
No	Ref	
Other NCDs		
Yes	0.91 (0.29–2.88)	0.88
No	Ref	
Obesity		
Yes	0.69 (0.31–1.54)	0.38
No	Ref	

HIV, human immunodeficiency virus; NCDs, non-communicable diseases; SARS-CoV-2; severe acute respiratory syndrome coronavirus 2.

## Data Availability

The dataset analysed in this study are available with the corresponding author upon reasonable written request.

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
