# Peer review of "Persistence of SARS-CoV-2 IgG Antibody Response among South African Adults: A Prospective Cohort Study"

_vaccines, 2023, doi:10.3390/vaccines11061068_

Round 1

Reviewer 1 Report

In the present scenario, investigating the aftermath effect of post-Covid- 19 induced antibody responses are specifically important in an African population. The current manuscript entitled Persistence of SARS-COV-2 IgG Antibody Response among South African Adults: A Prospective Cohort Study” by Adeneyi et al illustrated the durability of severe acute respiratory coronavirus-2 (SARS-CoV- 12 anti-nucleocapsid (anti-N) immunoglobulin G (IgG) after infection, and examines its association with established risk factors among South African healthcare workers (HCWs). In addition, the results of this study support the longevity of vaccine responses against SARS-CoV-2 in black Africans. However, some of the following concerns need to be addressed by the author.

1.) The main limitation to be addressed in the present study is in the baseline characteristic of participants (Table.1) is the COVID 19- infected participant received any vaccine dose or not? need to be addressed. If these participants receive the vaccine, the correlation between the vaccine and infected participants having IgG immunoglobulin G needs to discuss in the discussion.

2. From The same table. 1, BMI having more than 30 possesses 72.8 percent, so do authors have a comment on this?  Because the durability of immunoglobulin G (IgG) after Covid 19 infection might be depend on the BMI.

3. At 203 Sentences, the author mentioned, the mean extended durability of SARS-CoV-2 anti-N IgG antibodies was 223 days 203 (7.46 months). So in this line, the author needs to mention which table or figure he is mentioning like the 199th sentence.

4. In Table .4, why authors have not addressed or included TB or another respiratory disease? This needs to be addressed in the discussion.

5. At 49 th sentence the following recent studies need to cite to strengthen the statement (J Immunol. 2023 May 5:ji2200279. doi: 10.4049/jimmunol.2200279) and Cancers (Basel). 2022 Nov 17;14(22):5648. doi: 10.3390/cancers14225648.

6. Another limitation as the author mentioned in the limitations this is geographically precise and includes a relatively slight number of HCWs, potentially limiting generalizability to other settings or populations

7. In sentences (280-281) the author stated the current study is consistent with other studies so the authors need to refer to or cite the specific study.

Another major concern is the authors have not cited any recent works related to this work (later 2021 to 2023), so authors must cite  recent studies.

Minor English corrections are required 

Author Response

REVIEWER 1

Comments:

  1. The main limitation to be addressed in the present study is in the baseline characteristic of participants (Table.1) is the COVID 19- infected participant received any vaccine dose or not? need to be addressed. If these participants receive the vaccine, the correlation between the vaccine and infected participants having IgG immunoglobulin G needs to discuss in the discussion.

Response: Thank you for the comment. Data collection for the current study was carried out between  November and February 2021, before the COVID-19 Vaccines became available in South Africa.

  1. From the same table. 1, BMI having more than 30 possesses 72.8 percent, so do authors have a comment on this?  Because the durability of immunoglobulin G (IgG) after Covid 19 infection might depend on the BMI.

Response: Thanks for your valuable comment. There is high prevalence of obesity in the study population as shown in Table 1. However, on further sub-analysis, the classes of obesity; class 1 – 3 were not significant in both chi square test and logistic regression model and fails to  add additional value to our results. Hence, we kept the chose to keep the best model in the results (Tables 1 – 3).

  1. At 203 Sentences, the author mentioned, the mean extended durability of SARS-CoV-2 anti-N IgG antibodies was 223 days 203 (7.46 months). So, in this line, the author needs to mention which table or figure he is mentioning like the 199th

Response: Thank you for the comment. This result was provided as narrative report from our statistical analysis of the data.

  1. In Table 4, why authors have not addressed or included TB or another respiratory disease? This needs to be addressed in the discussion.

Response: Respiratory diseases were added as a variable. However, the number of participants who presented with respiratory diseases was too small for further analysis. This is highlighted in the study limitations.

  1. At 49 the sentence the following recent studies need to cite to strengthen the statement (J Immunol. 2023 May 5:ji2200279. doi: 10.4049/jimmunol.2200279) and Cancers (Basel). 2022 Nov 17;14(22):5648. doi: 10.3390/cancers14225648.

Response: Thank you for the suggestion. The suggested paper has been cited and added to our reference list.

  1. Another limitation as the author mentioned in the limitations this is geographically precise and includes a relatively slight number of HCWs, potentially limiting generalizability to other settings or populations.

Response: Thanks for your observation. This limitation cannot be ignored given the small number of non-black Africans in the study. Thus, future studies should explore all other racial groups to gain broader understanding of the durability of the humoral immune response (SARS-CoV-2 anti-N IgG antibodies) in the country.

  1. In sentences (280-281) the author stated the current study is consistent with other studies so the authors need to refer to or cite the specific study.

Response: Thank you for the comment. Supporting studies have been referenced.  

Another major concern is the authors have not cited any recent works related to this work (later 2021 to 2023), so authors must cite  recent studies.

Response: Thank you for the comment. We have updated our reference list to include recent studies.

Reviewer 2 Report

The manuscript presents the antibody response to SARS-CoV-2 infection in healthcare workers (HCW) in Africa and attempts to correlate the response with various factors such as age, race, diabetes, hypertension, obesity, HIV infection etc. the results are consistent with another study published from Tunisia in West Africa (PMID: 6658402).

The following concerns need to be addressed:

1) sample size is small for many factors so comparison is difficult. for example, whites are in single digit in the population included in the study.

2) data in table 2 can be shown in histograms with statistical comparisons

3) analysis of the Antibody negative among PCR positive group for risk factors  might be helpful. e.g. smokers may have less antibody or no antibody though they are PCR positive or vice versa.

Author Response

REVIEWER 2

The manuscript presents the antibody response to SARS-CoV-2 infection in healthcare workers (HCW) in Africa and attempts to correlate the response with various factors such as age, race, diabetes, hypertension, obesity, HIV infection etc. the results are consistent with another study published from Tunisia in West Africa (PMID: 6658402).

The following concerns need to be addressed:

Comments:

  • Sample size is small for many factors so comparison is difficult. for example, whites are in single digit in the population included in the study.

Response: Thanks for the comments. The sample included truly reflects the racial distribution of healthcare workers in the study setting, which is predominantly black Africans. Therefore, a better understanding of humoral immune response of other racial groups warrant further researches. This limitation is acknowledged and need for more studies have been suggested.  

  • Data in table 2 can be shown in histograms with statistical comparisons

Response: Thank you for the comment. A table appears as a better option of displaying the results shown in Table 2.

  • Analysis of the Antibody negative among PCR positive group for risk factors might be helpful. e.g., smokers may have less antibody or no antibody though they are PCR positive or vice versa.

Response: Thanks for the observation. Table 1 showed that only 11 and 9 individuals were active and former smokers (both were not significant and were excluded from further analysis). However, a brief description of the antibody trends have been included.

Round 2

Reviewer 2 Report

Authors have provided more detailed analysis of the data and extended the discussion and methods.